# Risk of disease and willingness to vaccinate in the United States: A population-based survey

**Bert Baumgaertner**[1]*, **Benjamin J. Ridenhour**[2], **Florian Justwan**[1], **Juliet E. Carlisle**[3], **Craig R. Miller**[4]

1 Department of Politics and Philosophy, University of Idaho, Moscow, Idaho, United States of America,
2 Department of Mathematics, University of Idaho, Moscow, Idaho, United States of America, 3 Department of Political Science, The University of Utah, Salt Lake City, Utah, United States of America, 4 Department of Biology, University of Idaho, Moscow, Idaho, United States of America

* bbaum@uidaho.edu

## Abstract

### Background

Vaccination complacency occurs when perceived risks of vaccine-preventable diseases are sufficiently low so that vaccination is no longer perceived as a necessary precaution. Disease outbreaks can once again increase perceptions of risk, thereby decrease vaccine complacency, and in turn decrease vaccine hesitancy. It is not well understood, however, how change in perceived risk translates into change in vaccine hesitancy.

We advance the concept of vaccine propensity, which relates a change in willingness to vaccinate with a change in perceived risk of infection—holding fixed other considerations such as vaccine confidence and convenience.

### Methods and findings

We used an original survey instrument that presents 7 vaccine-preventable "new" diseases to gather demographically diverse sample data from the United States in 2018 ($N$ = 2,411). Our survey was conducted online between January 25, 2018, and February 2, 2018, and was structured in 3 parts. First, we collected information concerning the places participants live and visit in a typical week. Second, participants were presented with one of 7 hypothetical disease outbreaks and asked how they would respond. Third, we collected sociodemographic information. The survey was designed to match population parameters in the US on 5 major dimensions: age, sex, income, race, and census region. We also were able to closely match education. The aggregate demographic details for study participants were a mean age of 43.80 years, 47% male and 53% female, 38.5% with a college degree, and 24% nonwhite. We found an overall change of at least 30% in proportion willing to vaccinate as risk of infection increases. When considering morbidity information, the proportion willing to vaccinate went from 0.476 (0.449–0.503) at 0 local cases of disease to 0.871 (0.852–0.888) at 100 local cases (upper and lower 95% confidence intervals). When considering mortality information, the proportion went from 0.526 (0.494–0.557) at 0 local cases of disease to 0.916 (0.897–0.931) at 100 local cases. In addition, we ffound that the risk of mortality invokes a larger proportion willing to vaccinate than mere morbidity ($P$ = 0.0002), that

**Data Availability Statement:** All relevant data are within the manuscript and its Supporting Information files.

**Funding:** Research reported in this publication was supported by the National Institute Of General Medical Sciences of the National Institutes of Health under Award Number P20GM104420 (BB, BJR, CRM). The funders had no role in study design, data collection and analysis, decision to publish, or preparation of the manuscript.

**Competing interests:** The authors have declared that no competing interests exist.

**Abbreviations:** CFR, Code of Federal Regulations; CR, completion rate; GEE, generalized estimating equation; IRB, Institutional Review Board; QIC, quasi-information criterion; SSI, Survey Sampling International (Dynata); STROBE, Strengthening the Reporting of Observational Studies in Epidemiology.

older populations are more willing than younger ($P<0.0001$), that the highest income bracket (>\$90,000) is more willing than all others ($P = 0.0001$), that men are more willing than women ($P = 0.0011$), and that the proportion willing to vaccinate is related to both ideology and the level of risk ($P = 0.004$). Limitations of this study include that it does not consider how other factors (such as social influence) interact with local case counts in people's vaccine decision-making, it cannot determine whether different degrees of severity in morbidity or mortality failed to be statistically significant because of survey design or because participants use heuristically driven decision-making that glosses over degrees, and the study does not capture the part of the US that is not online.

## Conclusions

In this study, we found that different degrees of risk (in terms of local cases of disease) correspond with different proportions of populations willing to vaccinate. We also identified several sociodemographic aspects of vaccine propensity.

Understanding how vaccine propensity is affected by sociodemographic factors is invaluable for predicting where outbreaks are more likely to occur and their expected size, even with the resulting cascade of changing vaccination rates and the respective feedback on potential outbreaks.

## Author summary

### Why was this study done?

- In the US, vaccine-preventable diseases have gone down (~1970–2000s), followed by a rise in vaccine hesitancy (~2000–2018).

- In places where disease outbreaks have occurred, vaccination rates have gone back up (~2000–2018).

- We conducted this survey to better understand how local cases of disease can influence people to vaccinate.

### What did the researchers do and find?

- We conducted a survey that presented participants with one of 7 new disease outbreaks, each with a different degree of morbidity or mortality.

- We asked participants how many local case counts it would take for them to vaccinate against that disease.

- The risk of mortality was associated with greater willingness to vaccinate in the presence of fewer case counts compared to the risk of morbidity.

- Likewise, older populations were more willing than younger, people with high incomes were more willing than all income levels, men were more willing than women, and our findings suggest a relationship between willingness to vaccinate and political ideology.

**What do these findings mean?**

- Part of people's decision to vaccinate is their risk of contracting the disease, and this assessment can vary across different populations.

- This information can be helpful for campaigns that aim to reduce vaccine hesitancy and is useful for modeling feedback between human decision-making and the spread of disease.

## Introduction

In order to understand the recent decline in vaccination rates and the increase of nonmedical vaccine exemptions, research on the formation of vaccine attitudes has been on the rise [1]. "Vaccine hesitancy" is an important concept that has emerged out of this research and unifies several considerations. Vaccine hesitancy is understood as the delay in acceptance or refusal of vaccination despite the availability of vaccination services [2]. More specifically, it encompasses 3 factors that contribute to the complex decision-making process for vaccination: complacency, confidence, and convenience.

Vaccination complacency occurs when the perceived risks of vaccine-preventable diseases are sufficiently low so that vaccination is no longer perceived as a necessary precaution. For example, after the first measles vaccine was licensed in 1963, the number of reported cases in the US dwindled from the hundreds of thousands per year to about 1,000–10,000 per year by the 1980s, to fewer than 1,000 per year by the year 2000, when it was declared eliminated [3]. Without the threat of contracting measles, and similarly other vaccine-preventable diseases, the success of vaccination programs decreases the incentive to vaccinate, thereby providing room for complacency and, consequently, vaccine hesitancy. Furthermore, a decrease in vaccine confidence and lack of convenience can further exacerbate vaccine hesitancy.

Evidence of increased vaccine hesitancy in the US can be seen in the rise of nonmedical vaccine exemptions and, relatedly, a decrease in vaccination rates [4]. In order to combat dropping vaccination, some states, e.g., California and Washington, have instituted policies that prohibit nonmedical exemptions. It is expected that such changes in policies will help increase vaccination rates. Of course, such policies may change in the future. Furthermore, many states in the US have not adopted such a strategy and continue to allow for nonmedical exemptions. Currently, 18 states allow nonmedical exemptions [4].

Even without institutional changes, however, there is evidence that suggests that disease outbreaks can once again increase perceptions of risk, thereby decrease vaccine complacency, and in turn decrease vaccine hesitancy. For example, following a measles outbreak of more than 16,000 cases and 75 deaths in California from 1988 to 1990, Dales and colleagues [5] found that the strongest vaccination response occurred where media coverage was highest and that the response decayed with both time and distance [6–8]. Similarly for pertussis: when a county in the US experienced a large pertussis outbreak, the proportion of unvaccinated children there decreased significantly [9]. In a poll conducted by the Harvard School of Public Health in September 2009, of the adults who said that they did not intend to get the pandemic influenza vaccine for themselves or their children, 60% also said that they would change their mind if other members of the community were sick or dying from A(H1N1)pdm2009 [10]. Finally, in a meta-analysis of 34 studies, Brewer and colleagues [11] found that both the

perceived likelihood of being harmed by a disease and the perceived severity of that harm are significant predictors of vaccine behavior.

In sum, complacency is a major contributing factor to vaccine hesitancy, but hesitancy can be overcome by increasing perceived risk (which decreases complacency). What is not well understood, however, is how change in perceived risk translates into change in vaccine hesitancy. In other words, we do not yet have a sufficiently rich understanding of "vaccine propensity," a concept introduced by Justwan and colleagues [12] that we further advance here.

We define vaccine propensity as a mapping from perceived risk of infection to reported willingness to vaccinate, holding fixed other considerations, such as vaccine confidence and convenience. By "perceived risk of infection," we mean the combination of the probability of contraction and the severity of the disease, as understood in terms of morbidity and mortality. We see the concept of vaccine propensity as enriching the concept of vaccine complacency by providing a dynamic mechanism of how hesitancy may change in response to changes in the landscape of risk.

Strictly speaking, vaccine propensity is an individual attribute and is expected to vary across individuals. That is, given some information about disease prevalence and severity, individuals subjectively determine (or "perceive") their risk of infection. Individuals can then report their (un)willingness to vaccinate if the perceived risk is above (below) some subjective threshold. Alternatively, the threshold itself can be probed by asking how much risk someone is willing to accept (e.g., in terms of disease prevalence) before they are willing to vaccinate (holding fixed other concrete information about, e.g., disease severity). While vaccine propensity is individual or subjective, it is helpful to translate it to the aggregate level. This can be done by estimating a function that outputs the cumulative proportion of a sampled population that reports a willingness to vaccinate for a given level of disease prevalence as input. In this way, the concept of vaccine propensity can also be understood and illustrated at a population level. This is particularly helpful for epidemiological considerations, wherein we may want to identify subpopulations that are more or less responsive to disease risk. That is, we can look at the increase of the cumulative proportion of a population that is willing to vaccinate as disease prevalence increases. In light of these considerations, and given that our analysis emphasizes the population level, we will use the phrase "vaccine propensity" to denote the aggregate version.

This paper has 2 goals. The first is to enrich our understanding of the role of complacency in vaccine hesitancy by determining how changes in vaccine hesitancy are associated with changes in risk (complacency). We do this by using the concept of vaccine propensity (see Fig 1). To make "risk" more concrete here, we use "number of local cases" as the primary determinant of probability of infection. The second goal of this paper is to determine how vaccine propensity is associated with sociodemographic factors. It is known that a variety of variables predict vaccine status or attitude. For example, research shows that vaccination rates vary with sociodemographic factors such as income, marital status, and age [13–15]. It has also been shown that, while the average rate of nonmedical exemption has increased from 1.5% to 3% across more than 6,000 schools in California from 2007 to 2013, there are many schools and regions with rates between 10% and 20%, and that white children attending private schools from families with higher incomes tend to have much higher exemption rates [16, 17]. In Texas, exemptions statewide rose from 10,000 to 45,000 in the 2007–2015 span [18], and of the 14 schools with exemption rates between 15% and 40%, 6 are clustered in the Austin area [19]. Similar vaccine refusal clustering has been documented in Washington State [20] and Michigan [21]. In fact, as of 2018, several "hotspots" of nonmedical exemptions have been documented across the US and appear to continue to grow [4]. While the true reason for variation in nonmedical exemptions and vaccination rates is expected to be nuanced and involve cultural dimensions, we can reasonably expect that sociodemographic factors at least roughly track some of the sources of variation.

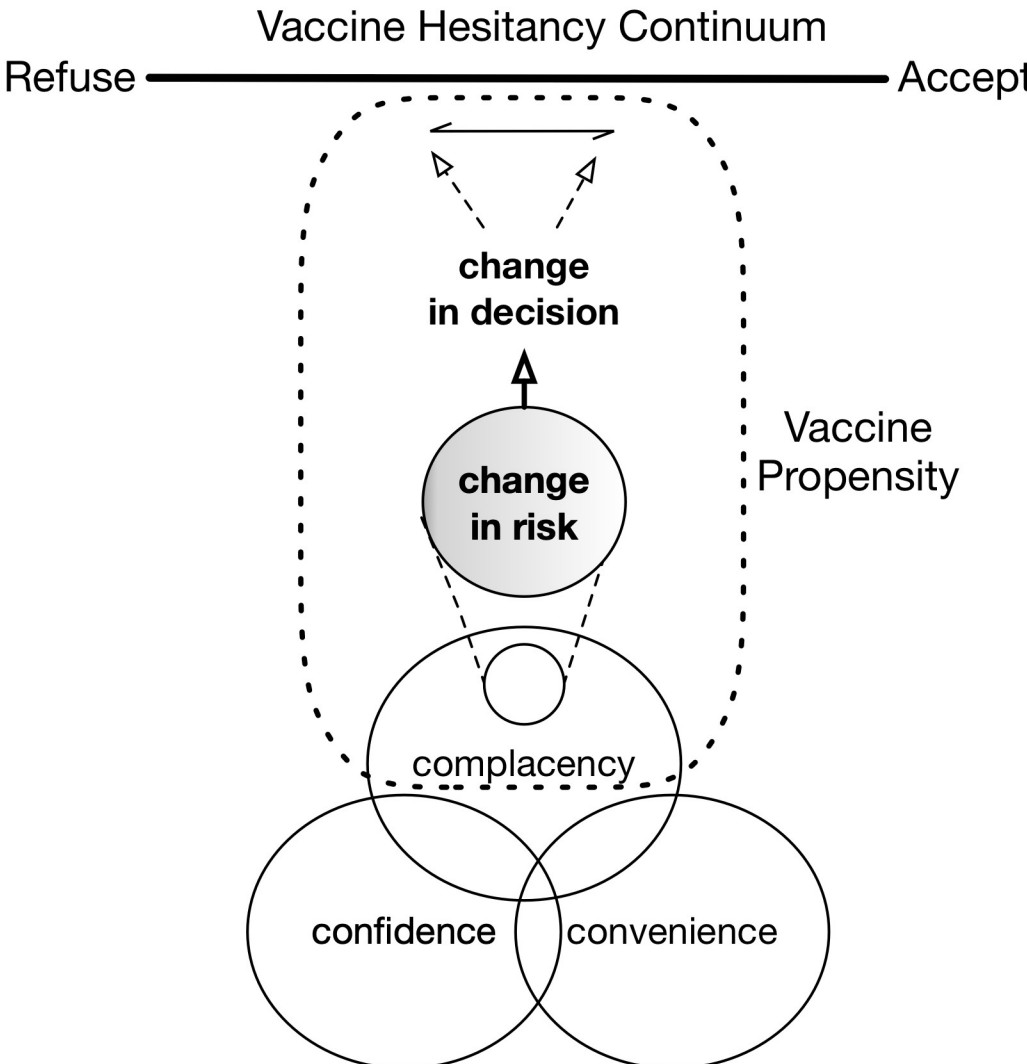

**Fig 1. Illustration of vaccine propensity.** The source of vaccine hesitancy has 3 components: complacency, confidence, and convenience. We are interested in understanding how a change in complacency due to a shift in risk—e.g., an increase in disease prevalence—maps onto a change in vaccine acceptance.

Both theory and empirical evidence indicate that the clustering of susceptible individuals makes outbreaks more probable [20–25]. It is not known, however, whether sociodemographic variables are also predictive of changes in vaccine behavior with respect to changes in risk of infection. If there are differences in vaccine propensity due to differences in sociodemographic makeup, we do not expect outbreaks to be equally probable, nor equally severe: some regions may be more responsive to changes in risk than others and thereby decrease the chances of a regional outbreak and/or decrease the size of an outbreak. Understanding how vaccine propensity is affected by sociodemographic factors is thus invaluable for predicting where outbreaks are more likely to occur and their expected size, even with the resulting cascade of changing vaccination rates and the respective feedback on potential outbreaks.

Furthermore, understanding sociodemographic factors of vaccine propensity would be useful for intervention strategies that make use of targeted messaging. It has been recognized that the "information deficit model" approach, which proceeds on the basis that vaccine hesitancy

is simply a matter of lack of knowledge, is not sufficient for changing vaccine behavior [26]. One way to improve communication strategies is to incorporate our understanding of the sociodemographic factors that influence hesitancy.

To help fill these knowledge gaps, we sought to better understand how vaccine hesitancy is associated with changes in risk (which we call vaccine propensity) and, in turn, how vaccine propensity is associated with common sociodemographic factors.

## Methods

This study is reported as per the Strengthening the Reporting of Observational Studies in Epidemiology (STROBE) guideline (S1 STROBE Checklist). Our study did not have a protocol or prespecified analysis plan with respect to empirically advancing the concept of vaccine propensity. The details of our analyses were decided upon after conducting the survey. Analyses were not data driven or exploratory—we opted for a model selection approach constrained by all our explanatory variables. Our knowledge-based model analysis was conducted in response to a request from a reviewer.

### Data collection and sample characteristics

For our statistical analysis, we rely on original micro-level data from a demographically diverse online survey programmed on Qualtrics (https://www.qualtrics.com). Participants for our online survey were collected via Survey Sampling International (SSI) (please note that SSI rebranded as Dynata after our study [https://www.dynata.com]). SSI is a US-based market research firm that maintains panels of persons used only for research. Panelists voluntarily join an SSI panel by responding to an online SSI advertisement (e.g., a banner advertisement on a website). SSI uses invitations of all types, including email invitations, phone alerts, and banners and messaging on panel community sites to include people with diverse motivations to take part in research. At the time of enrollment, new panelists are asked to join an online market research panel. At this point, it is made clear that it is not part of a sales process. Our survey invitations provide only basic links and information that is nonleading. Panelists are rewarded for taking part in surveys according to a structured incentive scheme, with the incentive amount offered for a survey determined by the length and content of the survey, the type of data being collected, the nature of the task, and the sample characteristics. Panelists are supported by a dedicated team and have the option to unsubscribe at any time. SSI's panel management is compliant with market research industry standards, data protection, and privacy laws.

Our quota sample (rather than a probability sample) includes individuals who were selected based on several demographic characteristics and resemble the US population according to 5 major dimensions: age, income, sex, race, and census region. Our sample was not designed to match population characteristics for education, but it still turned out to be relatively close in that regard. A detailed comparison of sample characteristics and cell percentages from the 2010 census can be found in Table 1.

Survey data were collected from January 25, 2018, to February 2, 2018. A total of 6,597 survey invitations were sent out by SSI. We had a total of 5,140 respondents that began the survey; 229 respondents discontinued the survey by the first "quality control" check, and another 32 discontinued by the second. The 2 "quality control" checks were the following 2 questions, respectively. Question 1 read as follows: "For quality control purposes, please select the number five with the letter 'G' next to it." Answer options were 5A, 5B, 5C, G, 5D, 5E, 5F, 5G, and 5H. Question 2 read as follows: "Research shows that people, when making decisions and answering questions, prefer not to pay attention and minimize their effort as much as possible.

**Table 1. Sample characteristics (compared to 2010 census).**

| Variable | Population (2010 Census) | Sample % (N) |
|---|---|---|
| **Age, y** | | |
| 18–24 | 13.08% | 16.0% (481) |
| 25–34 | 17.51% | 18.2% (547) |
| 35–44 | 17.51% | 18.1% (543) |
| 45–54 | 19.19% | 18.4% (553) |
| 55–64 | 15.55% | 14.5% (435) |
| 65 or older | 17.17% | 14.9% (446) |
| **Sex** | | |
| Male | 48.53% | 47.1% (1,404) |
| Female | 51.47% | 52.9% (1,574) |
| **Income (in USD)** | | |
| Less than $30,000 | 29.00% | 26.4% (792) |
| $30,000–$49,999 | 19.00% | 20.2% (607) |
| $50,000–$100,000 | 30.00% | 34.00% (1,023) |
| $100,000+ | 22.00% | 19.4% (583) |
| **Race/Ethnicity** | | |
| Hispanic or Latino | 16.30% | 14.1% (423) |
| White | 63.70% | 76% (2,335) |
| African American | 12.20% | 12.2% (395) |
| Asian | 4.70% | 4.4% (149) |
| **Region** | | |
| Northeast | 18.00% | 18.0% (541) |
| Midwest | 22.00% | 22.0% (661) |
| South | 37.00% | 37% (1,113) |
| West | 23.00% | 23.0% (691) |
| **Education** | | |
| Less than high school | 4.8% | 0.4% (13) |
| High school incomplete | 8.9% | 2.9% (87) |
| High school graduate | 31.0% | 19.9% (598) |
| Some college, no degree | 19.3% | 27.0% (812) |
| Two-year associate's degree | 8.6% | 11.2% (337) |
| Four-year college degree | 18.0% | 26.5% (798) |
| Postgraduate/professional degree | 9.3% | 12.0% (360) |

Our survey was designed to match population characteristics for age, sex, income, ethinicity, and region to data from the 2010 census. We include a comparison for education as well.

Some studies show that over 50% of people don't carefully read questions. If you are reading this question and have read all the other questions, please select the box marked 'other.' Thank you for participating and taking the time to read through the questions carefully! What is this study about?" Answer options were (1) Health, (2) Diseases, (3) Vaccines, and (4) Other. After deleting individuals who did not pass quality control checks, there were 3,007 valid responses. The completion rate (CR) for our survey is 58.5% and is based on the following calculation: the number of respondents who completed the survey divided by the number of those who began the survey, or CR = 3,007/5,140 = 58.5%.

The survey was structured as follows. First, respondents provided information about the size of their social network and the places they live in and visit during a typical week. Second,

participants read a brief description of a hypothetical infectious disease and then were asked a series of questions about how they would respond to this disease if it broke out in the US. The survey concluded with a third section in which respondents answered a series of questions about their political beliefs and basic demographic attributes. The survey questions for the variables in this paper are provided in S1 Text.

### Ethics statement

Before launching the survey, we obtained Institutional Review Board (IRB) exemption from the University of Idaho IRB (Project Number: 18–017; exemption granted under category 2 at 45 Code of Federal Regulations [CFR] 46.101[b][2]). Informed consent was obtained electronically from study participants before they began the online survey.

### Dependent variable measurement

Our dependent variable is "vaccine propensity," which we defined as the extent to which infection risk translates to a respondent's self-reported willingness to vaccinate. In order to measure this concept, we need to ascertain at what level of risk a given respondent would be willing to get vaccinated against a particular disease, holding other parameters constant. Given that individuals are likely to vary substantially in their knowledge about and experiences with any "real world" disease, we opted to rely on a hypothetical scenario.

Survey respondents were told that a new "disease has been discovered within the United States." We explained that any person who comes in contact with an infected individual has a 25% chance of contracting the disease. Providing this piece of information allowed us to hold the perceived infectivity of the condition constant. Next, we described that the disease causes a number of different symptoms: fever, diarrhea, vomiting, severe stomach pain, headaches and dizziness. Finally, in order to explore whether the severity of the disease influences vaccine propensity, we randomized the last bit of information across respondents: we split our sample into 7 even-sized groups and informed participants about the specific dangers associated with the infection. One group read that "people who contract the disease are sick for 1–3 days—too sick to work, go to school, care for others or leave the house." Three other groups read the same statement with the exception that the condition would last "4–7 days," "8–14 days," or "more than 15 days." The final 3 groups were informed that the illness could potentially be lethal and that it would kill approximately "1 in 1,000 people (0.1%)" who contract it, "1 in 100 people," or "1 in 10."

After reading one of these 7 different severity cues, respondents learned that a "highly protective vaccine (with minimal chances of side effects) is recommended and available" to them locally at no cost. Subsequently, all respondents answered a survey question that directly taps into vaccine propensity. In particular, we asked each respondent how many people in his/her local community would have to get infected with this disease for him/her to get vaccinated. Answer options were as follows: (1) no one, (2) 1, (3) 10, (4) 100, or (5) other [write in], (6) I will not vaccinate for this disease, and (7) I do not know. Only 45 answered "other," 395 answered "I do not know," and 160 did not answer the question. These individuals were omitted from the following analyses.

Thus, our final dependent variable has 5 distinct categories. Individuals who would vaccinate even if "no one" is infected and respondents who would not vaccinate for this disease at all are both unaffected by infection risk, but they occupy different end points on a spectrum. Individuals in our middle categories (1, 10, 100) are those survey respondents who display (varying levels of) sensitivity to infection risk—some of them requiring high numbers of locally reported cases before they would get vaccinated and some of them reacting to fairly low case

counts. Our statistical analyses assess the sociodemographic and risk factors that are associated with when individuals will seek vaccination.

## Independent variables, data processing, and analysis

All processing and analysis of the data was performed in R version 3.4.4 [27]. In order to predict vaccine propensity, we consider sociodemographic characteristics, including race, age, sex, income, education, population size of a respondent's hometown, population sizes of all cities commuted to during a typical week, number of children, age of youngest child (if applicable), political ideology, religious affiliation, religiosity (importance of religion and frequency of attending religious services), and the self-reported health of the respondent. All of the above variables were treated as factors with the exception of age, which was treated as a continuous variable. Some of the factors were treated as ordinal if an obvious ordering relationship exists, such as for income ranges and population sizes. We also controlled for whether the disease scenario presented to a given survey respondent involved mortality or morbidity. Only respondents who answered all of the questions pertaining to the variables above were included in our analyses (i.e., if any of the questions was not answered or answered as "unknown," then the observation was removed from the sample). The final sample size for analysis was 2,411.

Some of our independent variables were recoded prior to conducting the analyses. This was done since the case counts in some response categories were very low. The population sizes of cities commuted to during a week were reduced to the largest city commuted to, which was categorized (in order) as follows: does not commute, smaller than 1,000, 1,000–50,000, 50,000–250,000, 250,000–1,000,000, or more than 1,000,000. We used number of children and the age of the youngest child to compute a new variable that reflected whether the respondent had either no children, children at home (age of youngest child less than 19 years), or adult children. Political ideology is measured via 7-point Likert scale ranging from very liberal to very conservative. We omitted from our analyses respondents who identified as either "Libertarian" or "other." Respondents who identified themselves as either Jewish or Mormon were coded as "other," thereby providing us with the following categories for analysis: Protestant, Catholic, other, and not religious. Income level (originally recorded in US$10,000 intervals) was recoded as follows: less than $30,000, $30,000–$59,999, $60,000–$90,000, and more than $90,000. Respondents who answered that they had 1–8 years of education or did not complete high school were categorized as "no high school diploma"; those who graduated high school but either did not attend college or did not graduate from college were categorized as having "high school diploma"; those who either received a bachelor's degree, an associate's degree, or did not complete a graduate/professional degree were treated as "college graduates"; those who did complete a graduate/professional degree were treated as having "graduate education." Race was reduced to 3 categories: black, white, and other. The final number of respondents used for analysis was 2,411. Descriptive statistics are provided in Table 2.

Analysis of the survey data was conducted in the following manner. Our dependent variable was whether an individual would choose to get vaccinated (binary yes/no) at a particular risk level (0, 1, 10, 100 cases, or never). Thus our response variable was binary, and each individual had 4 repeated observations (one for each risk level less the "never" category). To control for the repeated observations, we utilized generalized estimating equations (GEEs) to fit a binomial model. We performed model selection in 2 phases: First, step-wise selection based on quasi-information criterion (QIC) was performed on all explanatory variables. Second, with the reduced set of explanatory variables, we then searched the model space where all pairwise interactions were considered (again, based on QIC). Model selection was halted after all remaining terms had a significance level ≤0.1. After model selection was completed, P values

**Table 2. Frequency of responses for survey questions that were used for analysis (information for age frequency can be found using S1 Code).**

| Question | Response | Percentage |
|---|---|---|
| **Local cases before vaccination** | | |
| | Always | 50 |
| | 1 | 18 |
| | 10 | 14 |
| | 100 | 7 |
| | Never | 11 |
| **Sex** | | |
| | Male | 49 |
| | Female | 51 |
| **Race** | | |
| | White | 77 |
| | Black | 12 |
| | Other | 11 |
| **Income (in USD)** | | |
| | <30,000 | 24 |
| | 30,000–60,000 | 29 |
| | 60,000–90,000 | 20 |
| | >90,000 | 27 |
| **Child status** | | |
| | No children | 37 |
| | At home | 34 |
| | Adult children | 29 |
| **Hometown population size** | | |
| | <1,000 | 8 |
| | 1 000–50 000 | 33 |
| | 50,000–250,000 | 31 |
| | 250,000–1,000 000 | 18 |
| | >1,000,000 | 11 |
| **Largest city population size commuted to** | | |
| | Do not commute | 53 |
| | <1,000 | 1 |
| | 1,000–50,000 | 11 |
| | 50,000–250,000 | 18 |
| | 250,000–1,000,000 | 12 |
| | >1,000,000 | 5 |
| **Education** | | |
| | No high school diploma | 2 |
| | High school diploma | 45 |
| | College graduate | 39 |
| | Graduate education | 13 |
| **Political leaning** | | |
| | Very liberal | 10 |
| | Liberal | 15 |
| | Slightly liberal | 10 |
| | Moderate | 28 |
| | Slightly conservative | 10 |

(*Continued*)

**Table 2.** (Continued)

| Question | Response | Percentage |
|---|---|---|
| | Conservative | 18 |
| | Very conservative | 8 |
| **Religion** | | |
| | Protestant | 27 |
| | Catholic | 22 |
| | Not religious | 24 |
| | Other | 27 |
| **Frequency of religious service attendance** | | |
| | Never | 27 |
| | Seldom | 24 |
| | A few times per year | 15 |
| | Once or twice per month | 9 |
| | Once per week | 20 |
| | More than once per week | 7 |
| **Importance of religion** | | |
| | Unimportant | 20 |
| | Not too important | 17 |
| | Somewhat important | 28 |
| | Very important | 34 |
| **Respondent health** | | |
| | Poor | 4 |
| | Fair | 21 |
| | Good | 56 |
| | Excellent | 18 |

Of the retained surveys, 58% ($N = 2{,}411$) presented mortality as a consequence of infection.

for independent variables were determined by ANOVA. In order to test the robustness of our analysis, we also used a knowledge-driven approach to building models. Our results are similar across our automatic model selection approach and the knowledge-driven approach. Details for the latter can be found in S2 Text.

## Results

The results of the analysis are presented in Tables 3 and 4. The step-wise model selection procedure determined that the best model (in terms of QIC) to predict vaccine propensity from our data used age, sex, political ideology, income, and the consequences of the disease (mortality versus morbidity); multiple interaction terms—all involving either age or risk (number of local cases)—were statistically significant.

We did not find significant relationships between vaccine propensity and the following variables ($P$ values reported from knowledge-driven approach, $P<0.01$ indicates statistical significance, see S2 Text for more details): religion ($P = 0.21$), religious importance ($P = 0.82$), frequency of attending religious service ($P = 0.8152$), hometown size ($P = 0.9397$), the size of the largest city commuted to ($P = 0.2909$), race ($P = 0.0209$), whether individuals had children or children at home ($P = 0.4771$), an individual's health ($P = 0.7275$), and the individual's educational attainment ($P = 0.0168$).

Fig 2 illustrates our concept of vaccine propensity and compares the morbidity and mortality scenarios. We can visually see how a change in vaccine propensity at the aggregate level

**Table 3. The resulting GEE model for vaccination propensity by respondent characteristics (N = 2,411).**

|  | Degrees of Freedom | $\chi^2$ | $P(>|\chi|)$ |
|---|---|---|---|
| **Political Leaning** | 6 | 19.27 | 0.0037 |
| **Scenario** | 1 | 13.52 | 0.0002 |
| **Age** | 1 | 21.14 | <0.0001 |
| **Sex** | 1 | 10.72 | 0.0011 |
| **Income** | 3 | 22.29 | 0.0001 |
| **Local Cases** | 3 | 1,190.05 | <0.0001 |
| **Political Leaning * Local Cases** | 18 | 37.93 | 0.0040 |
| **Scenario * Local Cases** | 3 | 6.25 | 0.1000 |
| **Age * Sex** | 1 | 3.43 | 0.0639 |
| **Age * Local Cases** | 3 | 23.69 | <0.0001 |
| **Age * Scenario** | 1 | 3.56 | 0.0591 |

Model selection was done based on QIC, and sandwich error variances were calculated to correct for individual effects. For the resulting model, pseudo-$R^2$ = 0.135 (see [41]).

**Abbreviations:** GEE, generalized estimating equation; QIC, quasi-information criterion

(understood here as the proportion of a population willing to seek vaccination, or "proportion seeking vaccination" for short) responds to a change in risk in terms of local cases. As risk increases in terms of the number of local cases of disease, so does the proportion seeking vaccination. In addition, the proportion seeking vaccination tends to be higher in the mortality scenario than the morbidity scenario for each risk level. In both scenarios, it is approximately 40%, a shift from 50% to 90%. That is, the difference between the proportion of the population willing to seek vaccination when risk is high (100 local cases) is around a 40-percentage-point increase from when risk is lowest (zero local cases).

Moreover, vaccine propensity appears to be gradual. That is, we do not observe that the 40-percentage-point increase occurs from one level of risk to the next, say from zero to 1 local

**Table 4. Pairwise comparisons of very liberal, moderate, and very conservative across our 4 levels of risk in terms of local cases (N = 2,411).**

| Contrast | Estimate | Standard Error | z | $P(>|z|)$ |
|---|---|---|---|---|
| **Local Cases = 0** | | | | |
| Very liberal–moderate | 0.4343 | 0.1547 | 2.808 | 0.0050 |
| Very liberal–very conservative | 0.4803 | 0.1988 | 2.415 | 0.0157 |
| Moderate–very conservative | 0.0460 | 0.1667 | 0.276 | 0.7827 |
| **Local Cases = 1** | | | | |
| Very liberal–moderate | 0.2611 | 0.1661 | 1.572 | 0.1161 |
| Very liberal–very conservative | 0.5855 | 0.2092 | 2.798 | 0.0051 |
| Moderate–very conservative | 0.3244 | 0.1731 | 1.875 | 0.0609 |
| **Local Cases = 10** | | | | |
| Very liberal–moderate | 0.2985 | 0.2028 | 1.472 | 0.1411 |
| Very liberal–very conservative | 0.8897 | 0.2410 | 3.691 | 0.0002 |
| Moderate–very conservative | 0.5912 | 0.1913 | 3.091 | 0.0020 |
| **Local Cases = 100** | | | | |
| Very liberal–moderate | 0.0648 | 0.2419 | 0.268 | 0.7887 |
| Very liberal–very conservative | 0.9144 | 0.2752 | 3.323 | 0.0009 |
| Moderate–very conservative | 0.8496 | 0.2189 | 3.882 | 0.0001 |

Results are given on the log odds ratio (not the response) scale.

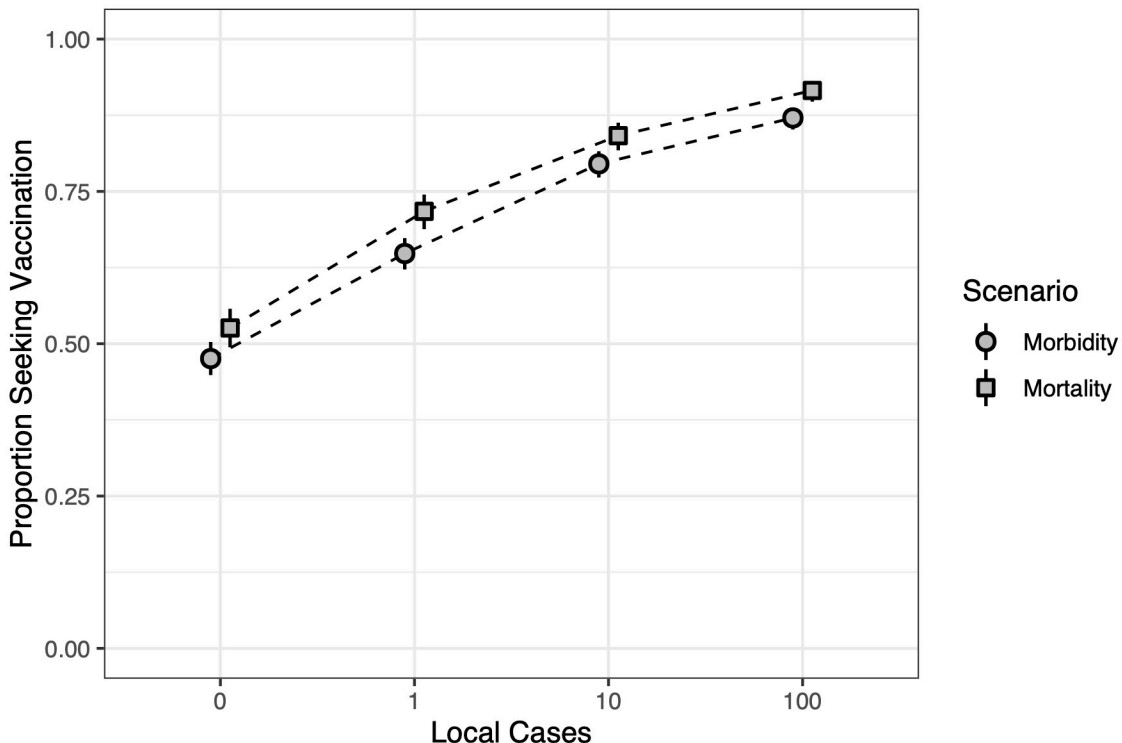

**Fig 2. The proportion of respondents willing to seek vaccination given number of local cases of disease for the morbidity and mortality scenarios.** The potential for mortality appears to be more strongly associated with willingness to vaccinate than morbidity. Zero local cases indicates that respondents are willing to vaccinate even if there are no local cases of the disease (though the disease does exist in the US). Slight offset in points and dotted lines are for visual aid.

case. Rather, we observe a more gradual increase over the 4 levels. This suggests that respondents are doing some, albeit rough, mental calculations regarding the probability of getting infected. And, given that the mortality scenario is higher than the morbidity scenario, this calculation is also taking into consideration the consequences of the disease.

One of the reasons for using the concept of vaccine propensity is to identify whether there are differential responses to risk levels for different types of populations. That is, we expect that some populations are more responsive to risk than others and that we would see this by comparing their changes in proportion willing to seek vaccination. In terms of our model, this means identifying sociodemographic variables that significantly interact with the variable we use to represent risk: number of local cases.

The variable for number of local cases had 2 interactions with sociodemographic variables (our scenario variable does not count as sociodemographic). The first is with political leaning (see Fig 3). Results suggest that a smaller proportion of respondents on the conservative end of the ideology spectrum are willing to seek vaccination than respondents who report being liberal. Specifically, at 0 local cases the proportion of "very conservative" respondents willing to vaccinate was 0.457 (0.386–0.528), and the proportion of "very liberal" respondents was 0.576 (0.511–0.638). At 100 local cases the proportion of "very conservative" respondents willing to vaccinate was 0.786 (0.721–0.840), and the proportion of "very liberal" respondents was 0.902 (0.860–0.932). Pairwise comparisons of very liberal, moderate, and very conservative respondents at each risk level can be found in Table 4.

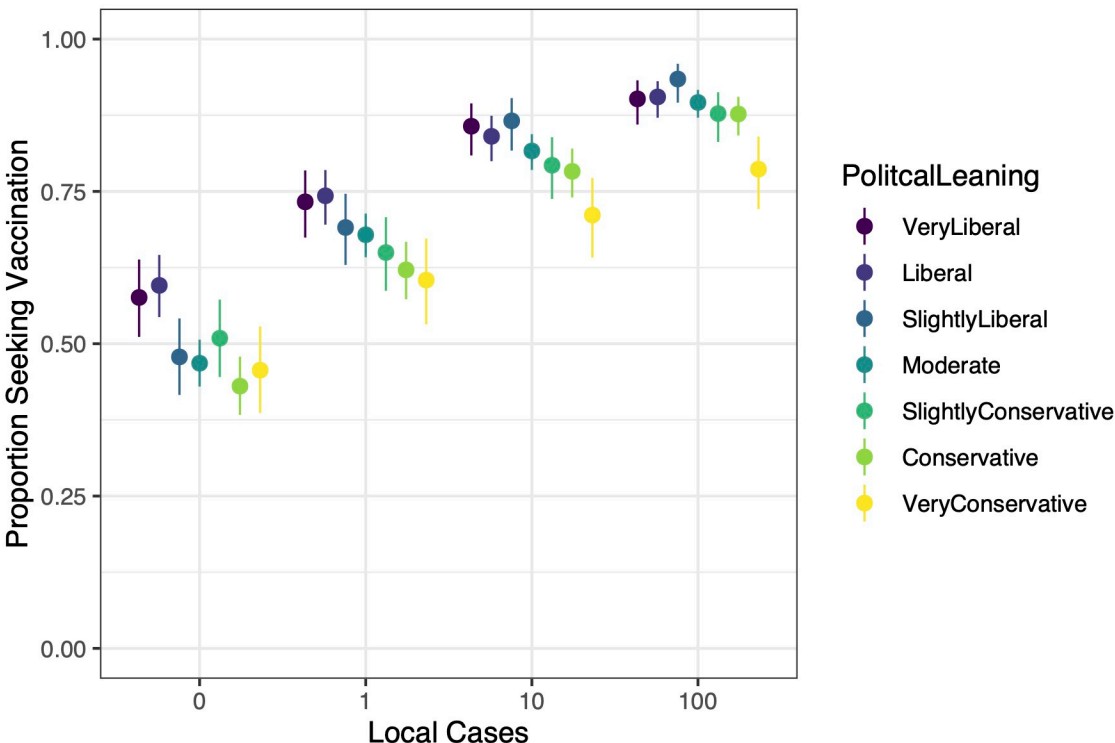

**Fig 3. Comparing ideologies in proportion willing to seek vaccination across 4 levels of risk.** Visual inspection suggests a rough trend that the conservative end of the ideology spectrum has a lower proportion willing to seek vaccination than the liberal end. Exact values can be found in Table 4. It is noteworthy to compare the very liberal, the very conservative, and the moderate. When risk is lowest at zero local cases, moderate respondents are similar to conservative and very conservative respondents. When risk is highest at 100 local cases, moderate respondents are similar to the liberal ends of the spectrum. The moderate population thereby exhibits a higher responsiveness to changes in risk. The points are offset from 0, 1, 10, and 100 as a visual aid. Whiskers from the points are the 95% confidence intervals.

One group that is particularly interesting to notice is "moderate." When the number of local cases is zero, we see that the proportion of moderates willing to seek vaccination is 0.468 (0.430–0.507), about as low as those that are conservative 0.430 (0.383–0.479) or very conservative 0.457 (0.386–0.528). When the number of local cases reaches 100, the proportion of moderates is 0.896 (0.871–0.917), just as high as the others, with the exception of respondents that are very conservative. In short, when risk is low, moderates respond much like very conservatives, but as risk increases moderates respond more and more like very liberals.

The other sociodemographic variable that risk interacted with is age ($P<.0001$). Fig 4 illustrates that risk assessments are being done differently across age groups for different risk levels. The proportion of younger respondents willing to vaccinate has the most variation, starting from 0.397 (0.360–0.436), when risk is at its lowest (0 local cases), going up to 0.900 (0.876–0.919), when risk is highest (100 local cases). As we consider older respondents, this variation decreases, with 0.653 (0.599–0.704) of the oldest respondents willing to seek vaccination at the lowest risk level and 0.878 (0.835–0.910) willing at the highest.

The proportion willing to vaccinate in the mortality scenario is nominally higher and less variable across ages than the morbidity scenario; however, the interaction between age and scenario did not reach statistical significance ($P = 0.0591$). Fig 5 illustrates the observed relationship between age and proportion willing to vaccinate for each scenario. The proportion willing to vaccinate appears relatively stable across age groups in the mortality scenario.

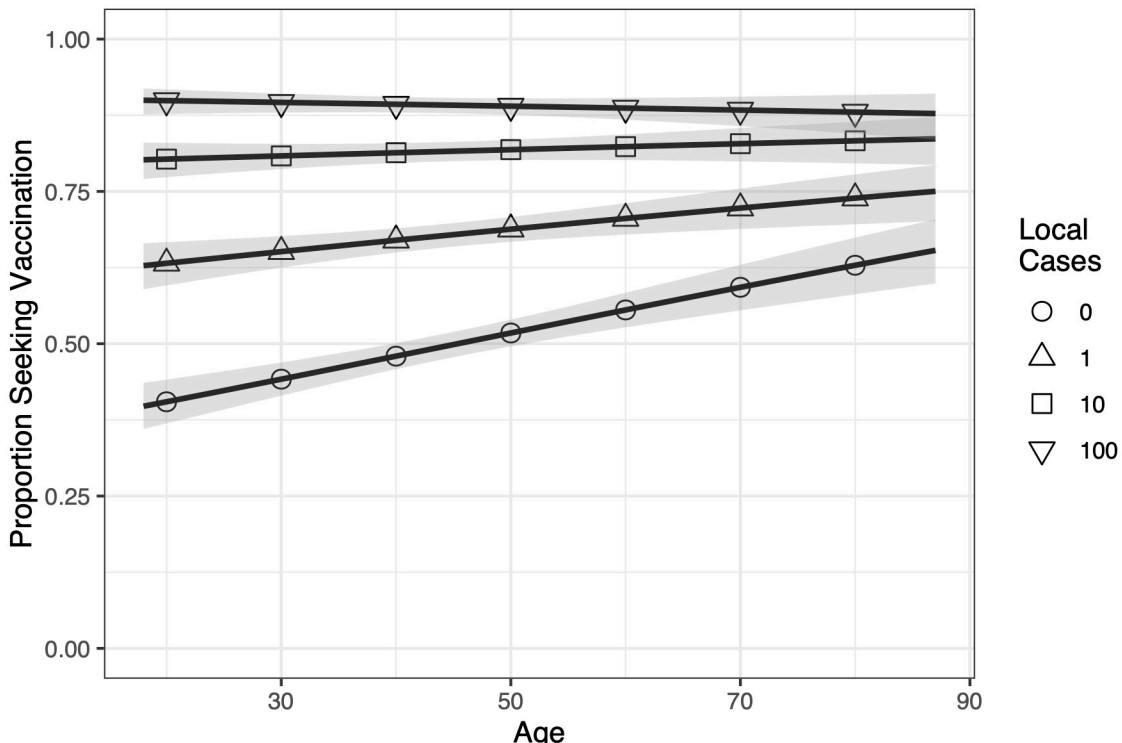

**Fig 4. For each level of risk, we plot the proportion of respondents willing to seek vaccination across ages.** Younger populations vary more in proportion willing to seek vaccination than older ones, i.e., when risk is lowest (0 local cases) the young have the lowest proportion, but when risk is highest (100 local cases) the proportion is at least as high for the young as the old. This illustrates a key idea behind vaccine propensity: some populations are more responsive to changes in risk than others. The solid lines are the trend lines, and the shaded area is the 95% prediction interval.

Moreover, older people seem to treat the mortality and morbidity scenarios similarly, as the proportion willing to vaccinate in both are nearly identical. In the morbidity scenario, however, fewer young people are willing to vaccinate than older people. However, it is important to note that these differences did not reach statistical significance.

Although the interaction between age and sex approached but did not reach statistical significance ($P = 0.0639$), Fig 6 illustrates our observation that for younger populations, similar proportions of men and women are willing to vaccinate (0.716 [0.671–0.757] and 0.716 [0.673–0.755], respectively, at 95% CI.) As age increases, a larger proportion of men (0.837 [0.792–0.874]) are willing to seek vaccination than women (0.739 [0.667–0.800]), although with overlapping confidence intervals. We provide a possible interpretation in the discussion section.

Finally, family incomes above $90,000 are willing to vaccinate in higher proportion at 0.801 (0.774–0.825) than those with incomes below that ($P = 0.0001$) (see S2 Text for comparisons in our knowledge-driven approach). Specifically, the proportion of families with incomes in the $60,000–$90,000 range was 0.718 (0.682–0.752), in the $30,000–$60,000 range was 0.735 (0.705–0.762), and in the below $30,000 range was 0.717 (0.683–0.750) (see Fig 7).

## Discussion

Our study objective was to better understand how vaccine hesitancy is associated with changes in risk (vaccine propensity) and, in turn, how vaccine propensity is associated with common

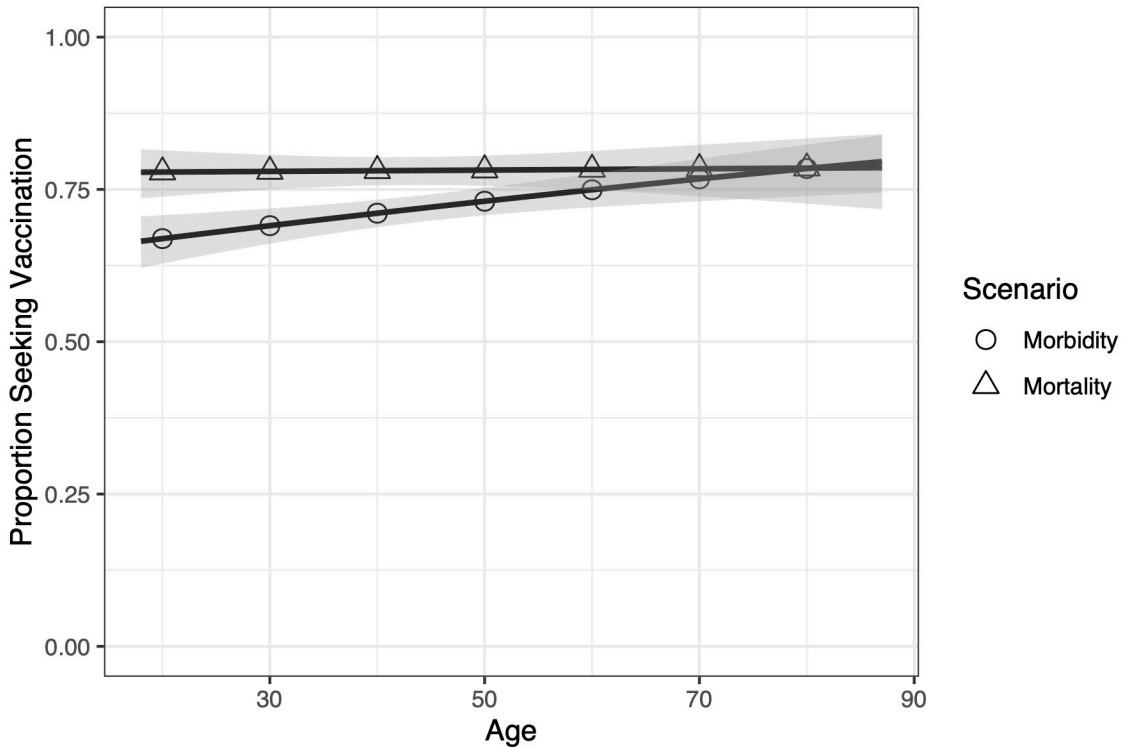

**Fig 5. Comparing the morbidity and mortality scenarios across ages.** The mortality scenario is relatively stable across ages, while the morbidity scenario changes. Younger populations are less motivated by morbidity than mortality, but older populations respond similarly to both scenarios. The solid lines are the trend lines, and the shaded area is the 95% prediction interval.

sociodemographic factors. To do this, we collected and analyzed original micro-level data from a demographically diverse online survey.

In brief, we find an overall change of at least 30% in proportion willing to vaccinate as risk of infection increases, where risk is understood in terms of number of local cases. In addition, we find that the risk of mortality invokes a larger proportion willing to vaccinate than mere morbidity, that older populations are more willing than younger, that the highest income bracket (>$90,000) is more willing than all others, that men are more willing than women, and that the proportion willing to vaccinate can depend on both ideology and the level of risk.

It is known that changes in risk correspond with changes in rates of vaccination and non-medical exemptions: decreases in risk can lead to increases in nonmedical exemptions and lower vaccination rates [16–18, 20, 21], while increases in risk can lead to decreases in non-medical exemptions and higher vaccination rates [5–11]. Moreover, it is known that vaccination rates vary with sociodemographic factors such as income, marital status, and age [13–15] and that vaccine hesitancy also has other sociodemographic determinants [2]. It is not known, however, how much of a change in risk is required to produce a change in vaccine hesitancy, nor do we know the sociodemographic variables that moderate these changes. Thus, the goals of this paper were to empirically advance the concept of vaccine propensity to enrich our understanding of complacency in vaccine hesitancy and to study the sociodemographic factors that are associated with vaccine propensity.

With respect to the first goal, we studied several scenarios of severity associated with disease, 4 ranging in different levels of morbidity and 3 in chances of mortality. For each scenario, we measured people's acceptable risk of infection in terms of the number of local cases of

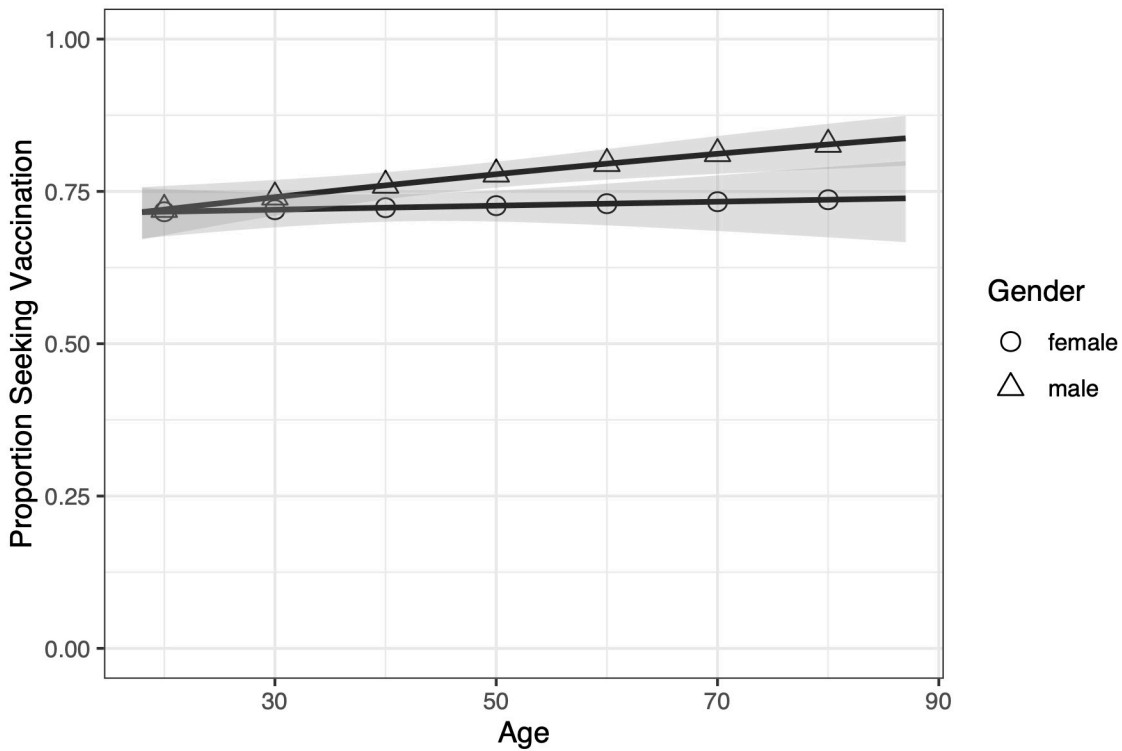

**Fig 6. Proportion of men versus women willing to seek vaccination across age groups in the morbidity scenario.** Men and women respond similarly when young, but as age increases a larger proportion of men will seek vaccination than women. The solid lines are the trend lines, and the shaded area is the 95% prediction interval.

infections that it would take for someone to get vaccinated. We were able to estimate a vaccine propensity relationship that started from the number of people that would vaccinate given zero cases of local infections. As the number of local cases of infection increased from zero to 1 to 10 to 100, the cumulative number of people that would vaccinate increased as well, with more people being added for lower increases of infection risk (e.g., from zero to 1) than for higher increases (e.g., from 10 to 100). Somewhat surprisingly, we did not find significant predictors within either morbidity or mortality individually, i.e., respondents did not seem to be motivated by, e.g., increases in how long symptoms last or how likely the infection is to cause mortality. However, we did find a significant difference between the morbidity scenarios and the mortality scenarios. In the morbidity scenario specifically, we estimated that around 48% would vaccinate at zero cases, 17% would vaccinate at 1 case, 15% would at 10 cases, and 7% at 100. Moreover, we found that respondents are more motivated by risk of mortality than risk of morbidity to vaccinate, which we intuitively expect—death is scarier than symptoms of disease. More specifically, we see the same rising trend in response to increasing risk in the mortality scenario as we saw in the morbidity scenario, but with an addition of about 5% more at each risk level.

These results give us a first estimate of vaccine propensity that provides a more detailed understanding of complacency. We can distinguish, e.g., not only between the heights of 2 vaccine propensity functions, which tells us which scenario or subpopulation has higher rates of reported willingness to vaccinate, but also between their slopes, which tells us about responsiveness to changes in disease prevalence. The vast majority of people would vaccinate for higher levels of infection risk (e.g., 100 or 10 cases), but as infection risk declines, so does

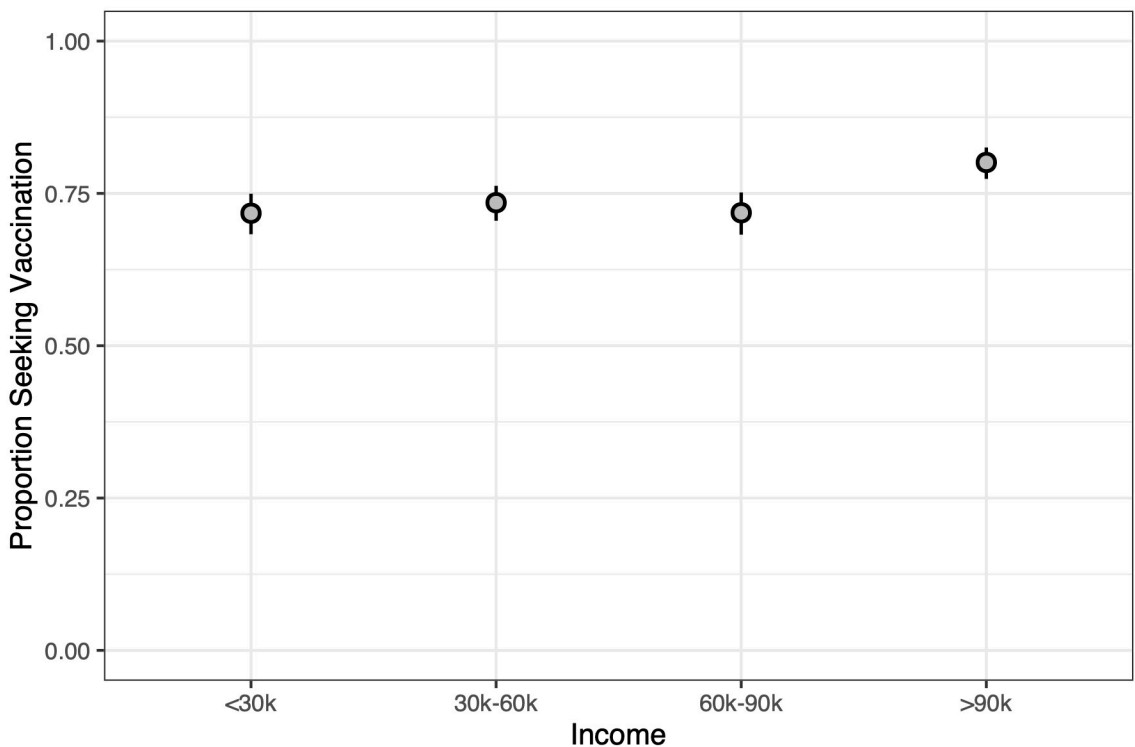

**Fig 7. Comparing different levels of income in their proportion to seek vaccination across ages.** Populations with family incomes above $90,000 are significantly greater in their proportion to seek vaccination than others, particularly in the 50-year-old and 60-year-old age groups. Whiskers from the points are the 95% confidence intervals.

intention to vaccinate, with the largest decrease in motivation seen when risk of infection goes from 1 to zero local cases. Put differently, we expect the largest increase in proportion willing to vaccinate to be when a single case of the disease first arrives locally, with a diminishing return as the number of local cases increases.

With respect to the second goal, we were able to identify several sociodemographic factors that are associated with vaccine propensity, including age, sex, income, and ideology. Given the extant empirical evidence on the relationship between age and risk [28, 29], we expected to see that younger populations would be less risk averse than older populations, with older respondents being more willing to vaccinate. Consistent with the literature, we found that older populations were more willing to vaccinate in order to mitigate risk of infection. Potential explanations for this include that older respondents are more likely to be concerned about mortality from infections and, additionally, are more likely than younger respondents to have experienced or witnessed diseases. However, the difference between younger and older respondents is not merely because the young are generally less averse to risk, since at high risk the young are just as averse. Rather, it seems that the young are more actively assessing risk against other factors—perhaps convenience—and making a calculated decision.

In terms of sex differences, the literature suggests women to be more risk averse than men [30]. However, contrary to the general literature on sex and risk, the association between vaccine propensity and sex was in the opposite direction in our study, with men being more likely to vaccinate than women. The only exception is for the youngest respondents, with similarly lower proportions willing to vaccinate in the 70%–75% range. One possible explanation for the difference between men and women is as follows. In a disease context, there are 2 risks being

weighed: risk of infection and risk of vaccine side effects. In our study, respondents were told that there were minimal chances of side effects. However, it is possible that the perceived severity of side effects is different across sex, with lower perceptions of severity among men and higher perceptions of severity among women. This explanation is consistent with a recent study in which women report more adverse effects of vaccination than do men [31]. It is also possible that men become more risk averse as they get older because their overall mortality risk is higher than women of the same age and this influences their decision-making.

With respect to income, Sakai [32] found that childhood vaccination rates at the country level rise and then fall with increases in income, with vaccination rates peaking around a per capita income of $30,000–$40,000. For the US counties specifically, 4 of the 7 vaccines examined also showed a peak in vaccination rates for middle-range incomes, with vaccination rates decreasing as incomes moved toward low and high. Similar results were obtained at the individual level, with the probability of a child being up to date on vaccination lower on both low- and high-income ends for many vaccines. We did not observe such a pattern when it comes to vaccine propensity: higher levels of income are associated with higher vaccine propensity. The difference could be that, in our case, we asked respondents to make a decision for themselves, while Sakai [32] focused on parents making decisions concerning their children. One purported explanation for why high-income parents have lower vaccination rates for their children is that high-income parents feel that they can protect their children through avoidance measures, thereby limiting their children's exposure to the threat of disease by reducing risk of infection [16, 33]. Moreover, higher-income respondents may have perceived the vaccination option described in our survey to be less costly than engaging in avoidance measures that would disrupt the routine places they visit (e.g., staying home from work). A second purported explanation for why childhood vaccination rates decline in high-income populations is that high-income parents may believe that they have better access to medical technology to treat or mitigate disease symptoms. Our survey instrument again does not present this as a viable possibility, since the only technological solution we present to respondents for the new disease is vaccination. Thus, while we cannot definitely rule out that respondents cannot "buy" mitigation, we did not encourage such thinking. Both of these purported explanations highlight health-related affordances made possible by higher levels of income that are alternatives to the option of getting a vaccination. Since our survey question includes vaccination as the only possible technological option, it may be tapping into a different pattern. However, since we cannot effectively test the range of technological options with our survey instrument, we refrain from drawing a specific explanation.

Given the college wage premium, it is reasonable to expect that the higher vaccine propensity of the >$90 income bracket would also mean a higher vaccine propensity in our education variable. Education was included in our survey instrument, but we found no significant relationship between vaccine propensity and education in our analysis. We therefore have reason to believe that whatever explanations are offered for our finding that the >$90,000 income bracket has the highest vaccine propensity are unlikely to be related to education.

Arguably our most interesting result pertains to political ideology. Strong conservatives make up a larger fraction of the vaccine-hesitant population than liberals [34]. Our results are consistent with these previous findings: the vaccine propensity of very conservative respondents is lower than that of very liberal respondents. Interestingly, however, if we focus on the slope of the propensity functions, we see a symmetric pattern related to ideology. While there are more very liberal than very conservative respondents who will vaccinate when there are zero local cases, increases in infection risk were associated with increased vaccine propensity for both groups. By contrast, visually the slope was greater among respondents who are moderate or less extreme ideologically, suggesting that they could be more motivated by increasing

infection risk. This is suggested by noting that when risk is lowest, those who are "middle of the road" ideologically are among the lowest proportion of those willing to vaccinate, but as risk increases to the highest level, they are among the highest willing to seek vaccination. This is consistent with findings that suggest that people with stronger or more extreme ideological views are less responsive to changes in the world than those with more moderate views [35]. In brief, the more entrenched or strong a person's ideological leaning, the more steadfast s/he will be in their (un)willingness to vaccinate in response to changes in risk. Moderates may be more pragmatic in their decision-making by focusing more on the risk of contracting the disease and focusing less on how that decision fits into other aspects of their worldview.

Limitations of our study are as follows. Evidence discussed in the introduction suggests that an increase in case counts can decrease vaccine hesitancy. Consequently, we focused on how local case counts were associated with vaccine decisions, holding fixed other considerations. Nevertheless, it is possible that case counts interact with or differentially weight other considerations that are not captured by our focus on local case counts alone (such as social influence). Moreover, our survey instrument captured differences between the morbidity and mortality scenarios but failed to detect significant differences within these categories. We are unable to say whether this is an artifact of our survey design or whether people's decision-making is heuristically driven and glosses over degrees in the mortality and morbidity categories. Other limitations concern sample size and representativeness. Our survey design ensured demographic diversity by matching the US census on age, sex, income, race/ethnicity, and region. After data collection, we also found that our sample closely matched for the education variable. However, it is possible that our sample differs from the population in other respects. For example, because our survey is online, it is unable to capture the part of the US that does not have access to the internet. We are also not able to capture those who did not complete the survey or failed our attention checks. Finally, as with any survey, it is possible that responses do not perfectly reflect actual traits or behaviors. Specifically to us, our survey asked participants to consider a counterfactual scenario, and it is possible that actual reactions to disease outbreaks would differ from the predictions that individuals made of themselves.

Our findings have important connections to epidemiological modeling and public health interventions. A wealth of work is being done in recognition that there is feedback between changes in human behavior in response to disease and the spread of disease [36–40]. By improving our understanding of how a change in risk of contracting a disease relates to a change in willingness to seek vaccination, we can improve epidemiological models often used to inform public health interventions. More specifically, we have advanced our understanding by (i) quantifying how different degrees of risk (in terms of local cases of disease) correspond with different proportions of populations willing to vaccinate and (ii) uncovering how those proportions will reflect populations that differ across sociodemographic variables, particularly those related to age, sex, income, and political ideology.

## Supporting information

**S1 STROBE Checklist. Completed STROBE Checklist.**
(PDF)

**S1 Text. Survey questions document.**
(PDF)

**S2 Text. Robustness check using knowledge-driven approach.**
(PDF)

**S1 Data. Collected data.**
(CSV)

**S1 Code. R script for data analysis.**
(R)

## Acknowledgments

We thank the Institute for Modeling Complex Interactions at the University of Idaho for its continued support of interdisciplinary endeavors.

Disclosure: The content is solely the responsibility of the authors and does not necessarily represent the official views of the National Institutes of Health.

## Author Contributions

**Conceptualization:** Bert Baumgaertner, Benjamin J. Ridenhour, Florian Justwan, Juliet E. Carlisle, Craig R. Miller.

**Data curation:** Benjamin J. Ridenhour, Florian Justwan.

**Formal analysis:** Benjamin J. Ridenhour, Craig R. Miller.

**Funding acquisition:** Bert Baumgaertner, Craig R. Miller.

**Investigation:** Bert Baumgaertner, Benjamin J. Ridenhour, Florian Justwan, Craig R. Miller.

**Methodology:** Bert Baumgaertner, Benjamin J. Ridenhour, Florian Justwan, Juliet E. Carlisle, Craig R. Miller.

**Project administration:** Bert Baumgaertner.

**Resources:** Bert Baumgaertner.

**Software:** Benjamin J. Ridenhour.

**Visualization:** Bert Baumgaertner, Benjamin J. Ridenhour.

**Writing – original draft:** Bert Baumgaertner, Benjamin J. Ridenhour, Florian Justwan.

**Writing – review & editing:** Bert Baumgaertner, Benjamin J. Ridenhour, Florian Justwan, Juliet E. Carlisle, Craig R. Miller.

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
