## [Decision Letter · Decision Letter 0]

11 Mar 2020

Dear Dr. Baumgaertner,

Thank you very much for submitting your manuscript "Vaccine Propensity: a representative qualitative survey measuring how changing risk of disease affects willingness to vaccinate in the United States." (PMEDICINE-D-19-03203) for consideration at PLOS Medicine. 

Your paper was discussed with an academic editor with relevant expertise and sent to independent reviewers, including a statistical reviewer. The reviews are appended at the bottom of this email and any accompanying reviewer attachments can be seen via the link below:

[LINK]

In light of these reviews, we will not be able to accept the manuscript for publication in the journal in its current form, but we would like to invite you to submit a revised version that fully addresses the reviewers' and editors' comments. You will appreciate that we cannot make a decision about publication until we have seen the revised manuscript and your response, and we expect to seek re-review by one or more of the reviewers. 

We hope to receive your revised manuscript by Apr 01 2020 11:59PM. Please email us (plosmedicine@plos.org) if you have any questions or concerns.

Please let me know if you have any questions. Otherwise, we look forward to receiving your revised manuscript in due course. 

Sincerely,

Richard Turner PhD, for Caitlin Moyer, Ph.D.

Associate Editor, PLOS Medicine

rturner@plos.org

You mention that you aim to "introduce" vaccine propensity in the present paper, but this has been mooted previously (PMID: 31461443). Please amend the presentation as appropriate. 

Please adapt the title to better match journal style. We suggest: "Risk of disease and willingness to vaccinate in the United States: a population-based survey".

Please expand the "methods and findings" subsection of your abstract, aiming to describe the methods and approaches used in additional detail in the early part of the subsection. 

We ask you to quote aggregate demographic details for study participants in the abstract. 

Please add a new final sentence to the "methods and findings" subsection of your abstract, which should summarize the study's main limitations. 

Please begin the "Conclusions" subsection with "In this study, we found that ..." or similar. 

After your abstract, we ask you to add a new and accessible "author summary" section in non-identical prose. You may find it helpful to consult one or two recent research papers in PLOS Medicine to get a sense of the preferred style.

Please streamline the "Introduction" section of your main text to reduce discussion of the Wakefield saga, for example. 

Early in the "Methods" section, please state whether the study had a protocol or prespecified analysis plan, and if so attach the relevant document(s) as a supplementary file. Please highlight analyses that were not prespecified.

Noting the "table sample vs census" in the supplementary material, we suggest that you include a table presenting the characteristics of the study participants, preferably including absolute numbers of participants, early in the "Results" section. Please rephrase "White Alone".

Please restructure the early part of the "Discussion" section of your main text: the first paragraph should consist mainly of a summary of the paper's findings. 

Also, we ask you add a discrete discussion paragraph on study limitations. 

Please substitute "sex" for "gender", where appropriate, throughout your paper. 

Please avoid using italics for emphasis.

In the reference list, please abbreviate journal names consistently (e.g., "PLoS Med." for reference 4). 

Noting reference 11, please ensure that all references have full access information. 

Noting "S3 file", we suggest removing the IP addresses and any other information which could be used to identify study participants. Please let me know if it would be helpful to discuss this further. 

Please adapt your attached SRQR checklist so that individual items are referred to by section (e.g., "Methods") and paragraph number rather than by page or line numbers, as the latter generally change in the event of publication. 

Comments from the reviewers:

*** Reviewer #1: 

The authors report the results from a single online experimental study examining willingness to vaccinate for a novel, hypothetical disease. The design allows in principle for exploration of both variations across demographics and political attitudes and variation over severity of disease (through a between-subjects manipulation of the description in to 7 groups, 3 with differing degrees of mortality and 4 with varying duration of non-life threatening illness) and likelihood of exposure (through a response question that incorporated information about local cases). The authors frame their inquiry as introducing a new concept ("vaccine propensity"). While there is some value to further explorations of correlates to vaccination intention / attitudes, I have both conceptual concerns about what is being measured here and substantial uncertainty about the significance of the findings. 

To start, I don't really see how "vaccine propensity" is a novel construct. Many authors (including some cited in this paper) have explored likelihood of vaccination as various individual and situational factors are changed (e.g., in discrete choice experimental structures). The present authors take a novel approach to measurement, but it remains unclear how readers should think about what is being measured here and how it will relate to real world behaviors.

Second, I have a conceptual problem with the design of this study. The authors "we asked each respondent how many people in his/her local community would have to get infected with this disease for him/her to get vaccinated" and then provided response options that listed different numbers of local cases. The structure of the question not only assumes that the proportion of people affected in their local community is a variable that ordinarily affects people's decisions, it implies directly that it should be. Thus, I see strong potential for a demand effect, in which respondents use the local case information because they are told to do so. There is also a huge potential for bias based on the set of response options provided (consistent with the literature on option set biases in survey design.) In other words, I am not confident that changes in the response option set would not change the results. Thus, what is measured is which respondents react to a particular set of local case count options, and the graphics are showing the increase in cumulative willingness to vaccinate as we add each group of responses. 

The authors collapse over their 7 scenarios into either mortality or morbidity, thereby ignoring the variance in degree of either outcome. This appears to be because the degree did not affect results (per the second paragraph of the discussion but never discussed or demonstrated in the results). Nonetheless that should be shown especially since it is interesting to consider how long duration morbidity compares to rare mortality. 

In terms of results, I'm left with very unclear takeaways. Mortality evokes slightly stronger reactions than morbidity (surprising no one, since a disease that kills is clearly more severe than one that does not). However we have no sense whether we should see this difference as larger or smaller than appropriate. More local cases evokes stronger vaccination interest, which is expected because local prevalence increases chances of exposure and because it serves as an indirect measure of disease scope. (I suspect many people are using this value as a measure of infectiousness, in replacement of the provided 25% rate.) The slope of this effect, however, is dependent on the hypothetical scenario context, the parameters used, and the option set presented, which limits how much readers can generalize from it to the real world. The age gradient suggests that older adults are just generally more likely to want to vaccinate regardless whereas younger adults are more situation dependent. This is a mildly interesting finding, consistent with the literature on differences in processing styles and decision making over the life course. 

I agree that the political ideology information probably the most interesting part of these findings, and a paper that focused primarily in this domain might be of more interest to me. 

Minor points:

The authors change their Y axis scaling from graphic to graphic, thereby distorting readers' perceptions of the magnitude of the changes shown. This is poor communication practice in general, but it is particularly pernicious in this context where the effect sizes are (a) only involving changes of roughly .1 and (b) variable across analyses.

I'm also concerned by the authors' reframing of the local case information as "risk." While there is a relationship between local cases and likelihood of exposure, "risk" is a broader construct that this. The authors should remain concrete in their descriptions: this is a manipulation of local cases. 

There is little to no discussion of limitations, which is a significant omission given the many challenges in trying to generalize these findings beyond this particular survey.

*** Reviewer #2: 

This paper investigates population willingness to consider vaccination using a set of vignette questions administered to an online sample of adults in the US, and variations across demographic subgroups. The paper is well written and well-reasoned. In this review, I focus on methodological aspects of the study and leave substantive concerns to vaccination experts. Specific questions for the authors include the following:

* How many survey respondents (and what %) failed the three quality control checks?

* What was the survey response rate, and how was it calculated? Most journals now require that the specific formula be reported, such as those provided by the American Association for Public Opinion Research.

* Was the final sample weighted and were those weights used for all or some of the analyses reported?

* How confident are the authors that their sample will respond to hypothetical threat questions in a manner similar to how they would respond to a real threat? Many survey research experts frown on hypothetical questions because they have been found not to be predictive of subsequent behavior. This is, at a minimum, a limitation that should be acknowledged if not more aggressively addressed.

* Actually, the paper does not include a discussion of study limitations. This needs to be corrected.

* The number of respondents answering that they did not know what the likelihood was that they would vaccinate (n=395) seems sufficiently large that it might be informative to examine the demographic variables associated with that uncertainty as well. This could probably easily be accomplished with the data at hand.

* Sample sizes should be provided for the models shown in Tables 2 & 3.

* The final supporting table, that compares the sample with benchmark data, does not include education, and this seems to be a serious omission. 

* Relatedly, another important limitation, not acknowledged, is that the study sample purports to be nationally representative, but provides no information about the sample frame for the online survey that was conducted. Given that many Americans do not access the internet, the burden of proof is on investigators to make a convincing case that the sample is indeed nationally representative, as the paper's title implies.

*** Reviewer #3: 

I confine my remarks to statistical aspects of this paper. The general approach is fine but I have several issues to resolve before I can recommend publication.

Line 199 -- How were ordinal values treated? (they are tricky)

Line 205 - How much data was "missing"? Just deleting it is not usually good, unless there is very little missing

Lines 217-219 Don't further clump income this way. This increases both type I and type II error.

Lines 225-227 Can it be right to treat others as having graduate education? Maybe it is, but this needs some explanation.

Line 236-238 Do not use stepwise selection. All the results will be wrong. P values are too low, standard errors are too small, parameter estimates are biased away from 0.... See e.g. Harrell *Regression Modeling Strategies*. Instead, substantive knowledge should be used to build models. If the authors insist on an automatic method, LASSO is much better than stepwise (but not that good).

Table 2 There is no real R^2 for logistic regression. There are various alternative measures, none of which are perfect. Which was used?

Peter Flom

***

[LINK]

---

## [Decision Letter · Decision Letter 1]

24 Apr 2020

Dear Dr. Baumgaertner,

Thank you very much for submitting your revised manuscript "Shifting risk of disease and willingness to vaccinate in the United States: a population-based survey." (PMEDICINE-D-19-03203R1) for consideration at PLOS Medicine. 

Your paper was re-evaluated by a senior editor and discussed among all the editors here. It was also discussed with an academic editor with relevant expertise, and re-reviewed by three reviewers, including a statistical reviewer. The reviews are appended at the bottom of this email and any accompanying reviewer attachments can be seen via the link below:

[LINK]

In light of these reviews, I am afraid that we will not be able to accept the manuscript for publication in the journal in its current form, but we would like to consider a further revised version that addresses the reviewers' and editors' comments. Obviously we cannot make any decision about publication until we have seen the revised manuscript and your response, and we plan to seek re-review by one or more of the reviewers. 

In particular, please be sure to address the reviewer concerns regarding the limitations of the SSI dataset and the lack of response rates. If it is not possible to provide the response rates, we request that you please devote considerable space in the discussion to this limitation and the implications, in addition to addressing the points raised by all the reviewers.

In revising the manuscript for further consideration, your revisions should address the specific points made by each reviewer and the editors. Please also check the guidelines for revised papers at http://journals.plos.org/plosmedicine/s/revising-your-manuscript for any that apply to your paper. 

In your rebuttal letter you should indicate your response to the reviewers' and editors' comments, the changes you have made in the manuscript, and include either an excerpt of the revised text or the location (eg: page and line number) where each change can be found. Please submit a clean version of the paper as the main article file; a version with changes marked should be uploaded as a marked up manuscript.

We expect to receive your revised manuscript by May 08 2020 11:59PM. Please email us (plosmedicine@plos.org) if you have any questions or concerns.

We look forward to receiving your revised manuscript. 

Sincerely,

Caitlin Moyer, Ph.D.

Associate Editor 

PLOS Medicine

plosmedicine.org

1.Title: Thank you for revising your title. However as your survey was cross-sectional and none of your results speak to any time course of changing attitudes, we feel that the use of the word “shifting” is not congruent with the study. Again, we suggest: "Risk of disease and willingness to vaccinate in the United States: a population-based survey".

2.Abstract: Methods and Findings: Please present the study methods before the study findings. In the limitations sentence, please clarify what is meant: “...it does not consider how other considerations interact with local case counts in people's vaccine decision making…” as it is not clear what is meant by ‘other considerations’

3.Author Summary: Thank you for including an author summary. Please structure the author summary using bullet points for separate ideas within the three sections, rather than writing in paragraph format. Please see our author guidelines for more information: https://journals.plos.org/plosmedicine/s/revising-your-manuscript#loc-author-summary

4.Methods: Line 118-119: It is not satisfactory to state that: “With respect representativeness, we outsourced the distribution of our survey to Survey Sampling International which utilizes proprietary methods.”- please provide details or a reference to the methods relating to the survey administration including response rates, and also see the comments of reviewer 2. If response rates are not available due to the limitations of SSI, please address the concerns of reviewer 2 and devote substantial space to describing how this impacts the interpretation of the study’s findings as a paragraph of the limitations section of the Discussion.

5.Discussion: Line 333-336 (and similarly at line 369): Please remove the suggestion of a causal relationship from this sentence, and throughout, we suggest the following change: “Thus, the goals of this paper were: to empirically advance the concept of vaccine propensity to enrich our understanding of complacency in vaccine hesitancy, and to study the sociodemographic factors that are associated with vaccine propensity."

6.References: Please use the "Vancouver" style for reference formatting, and see our website for other reference guidelines https://journals.plos.org/plosmedicine/s/submission-guidelines#loc-references

(for example, journal title is inconsistent in #4 and #18).

7. Table 1: The “Income” measure requires units.

8. Table 2: Income, hometown size, commute city size need units associated with the values presented.

9. Table 3: Please provide the abbreviation for “df” in the legend.

10. Table 4: Please provide the abbreviation for “SE” in the legend.

11. Figure 2: Please see reviewer 1’s comments regarding the axes of graphs, and please begin the y-axis at zero (in this case, the axis should display from 0 - 1.0). Please explain why the points are offset from the “1” “10” “100” lines, etc. Please describe the purpose of the dotted line connecting the points in the legend.

12. Figure 3: Please see reviewer 1’s comments regarding the graph axes (in this case, the axis should display from 0 - 1.0). Please explain why the points are offset from the “1” “10” “100” lines, etc. In the legend, please define the whiskers extending from the points.

13. Figure 4: Please see reviewer 1’s comments regarding the graph axes (in this case, the axis should display from 0 - 1.0). Please explain the nature of the solid lines in the legend- are these trendlines? Please explain the shaded regions in the figure legends. As these are difficult to see, we suggest using different colors.

14. Figure 5: Please see reviewer 1’s comments regarding the graph axes (in this case, the axis should display from 0 - 1.0). Please explain the nature of the solid lines in the legend- are these trendlines? Please explain the shaded regions in the figure legends. As these are difficult to see, we suggest using different colors.

15. Figure 6: Please see reviewer 1’s comments regarding the graph axes (in this case, the axis should display from 0 - 1.0). Please explain the nature of the solid lines in the legend- are these trendlines? Please explain the shaded regions in the figure legends. As these are difficult to see, we suggest using different colors.

16. Figure 7: Please see reviewer 1’s comments regarding the graph axes (in this case, the axis should display from 0 - 1.0). In the legend, please define the whiskers extending from the points.

17. Supporting information file S3: It appears that you have included data that may breach participant confidentiality. Please remove potentially identifying information from the file (e.g. IP address, county and zip codes, latitudes and longitudes, etc.)

Comments from the reviewers:

Reviewer #1: This manuscript is a revision of a paper I reviewed previously. The authors have addressed a number of the concerns raised previously about the framing of their work and its implications. I still find the paper less focused than I would prefer, but this is an editorial choice. 

One issue that I believe was insufficiently addressed, however, is axis scaling in their figures. The authors state: "We have now rescaled the axes on figure 2-4 to be the same; we have done likewise for figures 5-7. We have updated the figure descriptions to make clear the two scales we have selected.." Figures 2-4 are scaled from roughly .3 to 1.0 despite the fact that it is reporting a propensity. There is no logical reason to use this scale (vs. the full 0-1.0 range that a propensity can vary over) except to magnify slope, which is amplifying readers' perceptions of the effect. Figures 5-7 are scaled from roughly 0.6 to 0.95. Same issue, same effect, same problem, except even more so. Worse, Figure 4 is of the same type as Figures 5-7, but it is on a different scale. Thus, the slope represented in Figure 5 for morbidity (a range of roughly 0.15 over the age distribution) looks significantly steeper than the slope in Figure 4 for 1 local cases (a range of roughly 0.12-0.15) despite the fact they show numerically very similar levels of variation. This is clearly distorting perceptions of results. Let me be clearer than I was previously. In reporting propensity, I find it hard to justify any Y axis scaling other than 0-1.0.

I also have a few minor but important issues that need resolution:

1) Kudos to the other reviewer for flagging an issue I should have caught the last time around: the presentation of this sample from SSI as "nationally representative". I acknowledge that the authors have now put a note in their limitations on this issue. However, speaking as a frequent user of online samples, I _never_ describe an SSI sample or any other online sample (except maybe GfK) as "representative" of anything. It is demographically diverse in ways that mirror US Census demographics, yes, and the authors took specific steps to achieve this (with clear success). But it is not "representative" not only because of limitations on access to the internet but because it is an opt-in panel: SSI recruits people who want to take surveys, which is not everyone, and this is a characteristic that plausibly might relate to survey responses on this type of scenario. Thus, this is a minor but to me important point: I ask the authors to remove all descriptions of their sample as "nationally representative" and frame it instead as "demographically diverse" or similar.

2) It's not just that survey providers like SSI don't report response rates, it's that their methodology doesn't align to the concept. I believe SSI is no longer sending links to participants (which participants would either reply to or not) but instead basically encourages panel members to sign on to their account and then surveys are dynamically assigned to them moment by moment. This approach (or any like it) does not have a true denominator for use in determining a response rate. I would reword this to clarify more details of the sampling process in place at that time.

3) What does matter for online survey quality, however, is completion. The authors report that over 40% of their respondents failed quality control checks and were omitted from analyses. That is a huge number, and hence readers deserve a bit more detail on the character of those checks so as to judge the implications. In addition, it represents one more way in which the analytical sample is not representative (per comment above), since a representative sample would include at least some people who fail those attention checks.

Reviewer #2: Thank you for continuing to work on this paper, which has been considerably improved. There are several methodological questions regarding data collection, though, that remain inadequately addressed. These problems largely stem from the sample employed for the survey, how it was selected, what proportion of sampled respondents participated, and how the sample was weighted for analysis. Below, these concerns are summarized.

* Inability to report response rates suggests a poor methodology that more than anything seriously challenges any claims that the sample is representative of anything. I would encourage the authors to push back on the survey group they worked with to obtain and report a convincing rationale for this complete lack of transparency. My guess is that the only potential reason for being unable to provide a response rate is the use of a very poor convenience sampling methodology. My personal opinion is also that failure to provide response rate information is grounds for rejection. Appreciate that the sample's demographics have been compared with Census data. However, one key measure is not included in those comparisons: the percent of the population that does and does not have internet access. There is no question that the sample will be unrepresentative of this characteristic.

* Providing a discussion of limitations is appreciated and improves the paper. Lack of a response rate, which leaves the authors unable to even begin to address nonresponse bias concerns, is missing from this discussion, however. 

* Failure to properly weight the sample also detracts from claims that findings are representative. Even the most carefully designed nationwide probability surveys include statistical weights. That a nonprobability sample does not need to use weights because it is already "representative" is laughable. 

* Another relevant topic that remains unaddressed is the question of the frame from which the sample for the survey was selected. Exactly how do individuals get recruited into the SSI sample frame? It is almost certainly not a random process and the reader is left to guess or search the internet for relevant information. I did look at the SSI (now Dynata) web site and was unable to locate anything helpful.

Reviewer #3: I am checking "Proceed without recommendation" because I am a little confused.

The authors have addressed most of my concerns. Remaining issues:

I wrote: Lines 217-219 Don't further clump income this way. This increases both type I and type II error.

The authors responded: If we understand what the reviewer is saying here, we agree with them but are not sure what action to take. Aggregating responses changes the variance structure of the data. However, categorical response for things like income are already a substitute for a continuous variable; so getting at some sort of "true" relationship with income level isn't possible. We are therefore already limited with what we can infer from the data. By aggregating and changing the variance structure of the income classification (say into 10k increments), we are changing the hypothesis being tested. In other words, the variance structure is no longer appropriate for testing

hypotheses about the 10-classes of income, but it IS appropriate for testing hypotheses about the new 4-classes of income. Also, it is important to recognize that income is a continuous

variable and not one where people are forced to make subjective decisions about which category something should be in; i.e., if we had originally presented individuals with the 4

choices of income level, their responses should (in theory) exactly match the aggregated version of the more finely divided data.

My further response. Certainly changing the variable changes the variance structure. But .... I'm confused as to why that means the authors have to use income as a 4 category variable rather than a continuous one or a 10 category one. How was income recorded in the original data? Was it continuous (that would be unusual) or was it 10 category (in which case, I still don't see why they changed it). I feel like I am missing something here. Clearly the authors are proficient at statistics, but I don't quite understand their reasoning.

-----

Regarding model selection:

Sorry, but while it is true that model selection is valuable and that there are many ways to do it, the debate about stepwise is pretty much over: It doesn't work. The output is wrong.

I favor using substantive knowledge. The authors know more about the data and the models than any automatic algorithm does. But if they insist on using an algorithm, they should use one that attempts to adjust for the problems of stepwise. I like LASSO but I would be fine with other penaized methods (there IS debate about which of these methods is best). 

They are correct that stepwise is used a lot. But ... that's not really a reason to keep using it.

Peter Flom

[LINK]

---

## [Decision Letter · Decision Letter 2]

6 Aug 2020

Dear Dr. Baumgaertner,

Thank you very much for re-submitting your manuscript "Risk of disease and willingness to vaccinate in the United States: a population-based survey." (PMEDICINE-D-19-03203R2) for review by PLOS Medicine.

I apologize for the delay in getting back to you with our decision. I have discussed the paper with my colleagues and the academic editor and it was also seen again by three reviewers. I am pleased to say that provided the remaining editorial and production issues are dealt with we are planning to accept the paper for publication in the journal.

[LINK]

We look forward to receiving the revised manuscript by Aug 13 2020 11:59PM. 

Sincerely,

Caitlin Moyer, Ph.D.

Associate Editor 

PLOS Medicine

plosmedicine.org

Requests from Editors:

1.Please completely respond to the remaining points of reviewer 1; specifically:

 -Explicitly clarify whether or not individual study-specific email invitations were sent to potential respondents

- Describe the quality control checks used by SSI/Dynata, such as attention check questions, timing algorithms, so that the reader may judge the potential for bias from screen-out processes. 

-Separate those who voluntarily discontinue the survey vs. those who are screened out of the survey. 

Please note that the editors feel that it is appropriate for the knowledge-based model to remain in the supporting information given that this was a secondary analysis conducted in response to feedback during peer review.

2.Throughout manuscript: Your study is observational and therefore causality cannot be inferred. Please remove language that implies causality, such as “effect of” and similar. Refer to associations instead.

3.Abstract: Methods and Findings: Please quantify the main outcomes presented in the abstract, and include p values and 95% CIs associated with the result for: “We find an overall change in proportion willing to vaccinate of at least 30% as risk of infection increases.In addition, we find that the risk of mortality invokes a larger proportion willing to vaccinate than mere morbidity, that older populations are more willing than younger, that the highest income bracket (> $90k) is more willing than all others, that men are more willing than women, and that the proportion willing to vaccinate can depend on both ideology and the level of risk.”

4. Author summary: “Why was this study done?”: In the first bullet point, or where most appropriate, please clarify points relevant to your study such as the setting (United States) and the time frames (i.e. vaccine hesitancy on the rise, vaccination rates have gone down, preventable diseases have gone up, etc, please provide some sense of the relevant time period)

5.Author summary: “What did the researchers do and find?”: Please revise the third bullet point to: 

• Likewise, older populations are more willing than younger, people with high incomes are more willing than all income levels, men are more willing than women, and our findings suggest a relationship between willingness to vaccinate and political ideology.

6.Introduction: Line 72-74: Please revise to: “The first is to enrich our understanding of the role of complacency in vaccine hesitancy by determining how changes in vaccine hesitancy are associated with changes in risk (complacency).”

7.Introduction: Line 76-77: Please revise to: “The second goal of this paper is to determine how vaccine propensity is associated with sociodemographic factors.”

8.Introduction: Line 111-112: Please remove “Specifically, we find that factors such as age, sex, income, and ideological leanings influence reported vaccine responsiveness to changes in risk.” as this statement implies causality. Please remove any study results and findings from the Introduction and conclude the Introduction with a clear description of the study question or hypothesis (please re-organize the Introduction if necessary).

9. Introduction: Lines 110-112: Please delete the sentences mentioning the findings from the Introduction, and instead conclude the Introduction with a clear description of the study question or objective. “In this paper, we provide relevant insights on factors that shape vaccine propensity. Specifically, we find that factors such as age, sex, income, and ideological leanings influence reported vaccine responsiveness to changes in risk.”

10.Methods: Please update where appropriate to indicate that SSI is now called Dynata.

11.Methods: Line 114: Please spell out “Standards for Reporting Qualitative Research (SRQR) checklist” at first use. However, we do not feel that this is the most appropriate checklist for your study. If you agree, please instead add the following statement, or similar, to the Methods: "This study is reported as per the Strengthening the Reporting of Observational Studies in Epidemiology (STROBE) guideline (S1 Checklist)."

12. Methods: Line 115-117: You state that there was no prespecified analysis plan, but please explicitly state when the analyses were decided upon; e.g., which analyses were determined before seeing the data and if any were data-driven or conducted in response to a reviewer request.

13.Methods: Line 122: Please replace "subject" with participant, respondent, individual, or person.

14.Methods 132-133: As requested by the reviewer, please describe the quality control selection process.

15.Methods: 136-38: In a subsection of the Methods, please note your exemption from IRB approval and add a description of how informed consent was obtained from study participants, whether written or oral.

16.Methods: Line 150 and Line 183: Please replace "subject" with participant, respondent, individual, or person.

17.Methods: 186-188: Please revise to: “Our statistical analyses assess the socio-demographic and risk factors that are associated with when individuals will seek vaccination.”

18.Methods: Line 234: Please define the abbreviation QIC at first use

19.Results: Throughout, please present findings in the text with 95% CIs and p values.

20.Results: Lines 248-252: Please revise to: “We did not find significant relationships between vaccine propensity and the following variables: religion, religious importance, frequency of attending religious service, hometown size, the size of the largest city commuted to, race, whether individuals had children (or children at home), an individual's health, and the individual's educational attainment.”

21.Results: Line 260-261: Please revise to “In both scenarios, it is approximately 40%, a shift from 50% to 90%.”

22.Results: Line 279-282: Please provide the proportions, 95% CI and p values to accompany this result: “Results suggest that a smaller proportion of respondents on the conservative end of the ideology spectrum are willing to seek vaccination than respondents who report being liberal|compare specifically \\very liberal" and \\very conservative." Exact values can be found in Table 4.”

23.Results: Line 292: Please present this result with 95% CIs and p values “The other sociodemographic variable that risk interacted with is age.”

24.Results: Line 315: Please present this result with 95% CIs and p values: “Our age variable also interacted with sex.”

25.Results: Line 320-321: Please present this result with 95% CIs and p values: “Finally, we had one main effect that did not significantly interact with other variables: income. Family incomes above $90 000 are willing to vaccinate in higher proportion than those with incomes below that”

26.Discussion: Please begin the Discussion with 1-2 sentences briefly summarizing what was done to address the study objectives.

27.Discussion: Line 388-389: Please revise to: “However, contrary to the general literature on sex and risk, the association between vaccine propensity and sex was in the opposite direction in our study, with men being more likely to vaccinate than women.”

28.Discussion: Line 432-433: Please revise to: “Education was included in our survey instrument, but we found no significant relationship between vaccine propensity and education in our analysis.”

29.Discussion: Line 436: - Please make sure that there are separate references explicitly supporting the statements that “liberals are less likely to vaccinate as part of their emphasis on natural lifestyles” and that “strong conservatives make up a larger fraction of the vaccine hesitant population” unless reference 34 supports all aspects of the claims of this sentence:

“While it may be the case that some liberals are less likely to vaccinate as part of their emphasis on “natural” lifestyles, strong conservatives make up a larger fraction of the vaccine hesitant population [34].”

30.Discussion: Line 442-446: Please revise to: “While there are more very liberal than very conservative respondents who will vaccinate when there are zero local cases, increases in infection risk were associated with greater vaccine propensity for both groups. By contrast, the slope was greater among respondents who are moderate or less extreme ideologically, suggesting they could be more motivated by increasing infection risk.” or similar.

31.Discussion: Line 457-458: Please revise to: “Consequently, we focused on how local case counts were associated with vaccine decisions, holding fixed other considerations.”

32.Discussion: Line 474: Please replace "subject" with participant, respondent, individual, or person.

33.Supporting information S1 file: SPQR Checklist: This is not the most appropriate checklist for your study, as your study is not qualitative (please report your study according to the relevant guideline, which can be found here: http://www.equator-network.org/); we suggest you report your study according to the STROBE guideline, and include the completed STROBE checklist as Supporting Information. When completing the checklist, please use section and paragraph numbers, rather than page numbers. Please add the following statement, or similar, to the Methods: "This study is reported as per the Strengthening the Reporting of Observational Studies in Epidemiology (STROBE) guideline (S1 Checklist)."

34.Table 1: Please indicate that the right-most column represents % (number)

35.Table 2: Under the importance of religion measure, there may be a typo in the category of “No too important”

36.Table 3: Please fully define abbreviations for GEE and QIC in the legend.

37.Figure 1: Figure 1 is missing. We will need to evaluate the final version of the figure.

38.Figure 3: We suggest changing the colors of some of the markers to make them easier to distinguish from each other.

Comments from Reviewers:

Reviewer #1: The authors have responded appropriately to my previous concerns about the figures. 

The main issues remaining from my perspective concern (a) the back and forth about the sample process and characteristics and (b) the modeling issues raised by Reviewer 3.

Re modeling. Stepwise is indeed problematic. Saying that results are very similar regardless of method used is good. But if so, why prioritize stepwise or frankly any algorithmic approach in the paper? I'm wondering whether the best choice is to essentially flip positions: make the main presentation the knowledge-driven approach, move the algorithmic model to the supplementary file. But I'll defer to others on this.

Re: sample. First, the authors should note that SSI is now Dynata. Second, I am somewhat confused and concerned, since the additional methods text both disagrees with what i thought SSI/Dynata is doing at present and is incomplete in some important ways. The authors need to confirm:

- Individual study-specific email invitations were sent to potential respondents? While this was their method many years ago, I had been under the impression that Dynata was moving to inviting people to log on to the survey portal to take surveys in general and that the specific survey assignment was not done until login. I'm skeptical of straight email recruitment, because response rates to email invitations for surveys have never been good (often <10%), yet these data suggest a very high response rate. But, regardless, confirming this is key.

- What are the quality control checks? These remain undefined, and they are screening out a huge fraction of people. SSI/Dynata needs to disclose to the authors, and the authors need to clearly summarize, what attention check questions, timing algorithms, etc. are being used to screen people out. Whatever sampling biases may exist between SSI/Dynata's population and the general population pale in comparison to the degree of potential bias risk from a screenout process that removes almost half of respondents. We need to know what is done in detail to assess the risk of bias. I think the results are of interest almost regardless of the answer, but we need the answer.

- The authors need to separate voluntary failures to complete from failing control checks. At the moment, we cannot tell how many people are choosing to stop taking the survey themselves vs. are being screened out.

Reviewer #2: Thank you to the authors for continuing to work on this manuscript. I am satisfied that they have addressed, as much as is possible, my earlier concerns, and have no additional recommendations to make.

Reviewer #3: The authors have addressed my concerns and I now recommend publication

Peter Flom

[LINK]

---

## [Editor Report · Decision Letter 3]

11 Sep 2020

Dear Dr. Baumgaertner, 

On behalf of my colleagues and the academic editor, Dr. Sanjay Basu, I am delighted to inform you that your manuscript entitled "Risk of disease and willingness to vaccinate in the United States: a population-based survey." (PMEDICINE-D-19-03203R3) has been accepted for publication in PLOS Medicine. 

PRODUCTION PROCESS

PRESS

PROFILE INFORMATION

Thank you again for submitting the manuscript to PLOS Medicine. We look forward to publishing it. 

Best wishes, 

Caitlin Moyer, Ph.D.

Associate Editor 

PLOS Medicine

plosmedicine.org